# The Prospects of the Two-Day Cardiopulmonary Exercise Test (CPET) in ME/CFS Patients: A Meta-Analysis

**DOI:** 10.3390/jcm9124040

**Published:** 2020-12-14

**Authors:** Eun-Jin Lim, Eun-Bum Kang, Eun-Su Jang, Chang-Gue Son

**Affiliations:** 1Department of Korean Medicine, Institute of Bioscience and Integrative Medicine, Daejeon University, 62 Daehak-ro, Dong-gu, Daejeon 34520, Korea; eunjinlimsydney@gmail.com (E.-J.L.); jang.ensu2@gmail.com (E.-S.J.); 2Department of Health and Exercise Management, Daejeon University, 62 Daehak-ro, Dong-gu, Daejeon 34520, Korea; kbume23@naver.com

**Keywords:** cardiopulmonary exercise test, chronic fatigue syndrome, myalgic encephalomyelitis, postexertional malaise

## Abstract

Background: The diagnosis of myalgic encephalomyelitis/chronic fatigue syndrome (ME/CFS) is problematic due to the lack of established objective measurements. Postexertional malaise (PEM) is a hallmark of ME/CFS, and the two-day cardiopulmonary exercise test (CPET) has been tested as a tool to assess functional impairment in ME/CFS patients. This study aimed to estimate the potential of the CPET. Methods: We reviewed studies of the two-day CPET and meta-analyzed the differences between ME/CFS patients and controls regarding four parameters: volume of oxygen consumption and level of workload at peak (VO_2peak_, Workload_peak_) and at ventilatory threshold (VO_2_@VT, Workload@VT). Results: The overall mean values of all parameters were lower on the 2nd day of the CPET than the 1st in ME/CFS patients, while it increased in the controls. From the meta-analysis, the difference between patients and controls was highly significant at Workload@VT (overall mean: −10.8 at Test 1 vs. −33.0 at Test 2, *p* < 0.05), which may reflect present the functional impairment associated with PEM. Conclusions: Our results show the potential of the two-day CPET to serve as an objective assessment of PEM in ME/CFS patients. Further clinical trials are required to validate this tool compared to other fatigue-inducing disorders, including depression, using well-designed large-scale studies.

## 1. Introduction 

Myalgic encephalomyelitis/chronic fatigue syndrome (ME/CFS) is a debilitating multisystem disease that affects more than 20 million people of all ages and races worldwide [1,2]. The core symptoms are severe fatigue, unrefreshing sleep, postexertional malaise (PEM), and cognitive dysfunction for more than 6 months [3]. Approximately 30% of the patients are housebound, and 50% are unable to work full time [4]. Despite the seriousness of the illness, ME/CFS has no established pathophysiology or diagnostic tests yet [4]. Moreover, the diagnosis of ME/CFS is often confused with other fatigue-related or comorbid illnesses (e.g., depression) due to the overlapping symptoms [5]. 

Although the clinical presentation of ME/CFS can be ambiguous, a 2015 report by the Institute of Medicine (IOM) states that PEM is a hallmark of this disease and helps distinguish it from other conditions [3]. PEM consists of exertional intolerance and worsening of symptoms following minor physical or cognitive exertion, which can be severe enough to leave the patients bedridden [6,7]. Studies on the prevalence of PEM in ME/CFS reported that more than 80% of all ME/CFS patients had experienced this symptom at some point in their illness [8,9]. The underlying pathophysiology of PEM is not clearly understood, but researchers have adapted the two-day cardiopulmonary exercise test (CPET) as a strategy for objectively measuring PEM [3,10,11,12].

The CPET was developed as a tool to measure the functional capacity of the body through analysis of gas exchange (e.g., oxygen, and carbon dioxide) during exercise for athletes and people with cardiac, pulmonary, vascular, and metabolic disorders [13]. In general, the CPET is performed on a single day for the above purposes; however, the two-day CPET has received attention particularly for ME/CFS since 2007 [10]. It can assess recovery capacity using two exercise tests administered 24 h apart [14]. A number of one-day CPET studies reported a lack of significant differences between ME/CFS patients and controls [15,16,17,18]; however, significantly different responses were observed in the 2nd test of the two-day CPET [10,19,20]. This may indicate impaired recovery, reduced energy production, and likely PEM in ME/CFS patients [14]. Multiple studies have reported an association with mitochondrial dysfunction as a potential pathophysiology in ME/CFS patients [21,22].

A previous review study assessed one parameter (the volume of oxygen consumption at peak, VO_2peak_) between ME/CFS patients and controls by combining the two different tests (one-day and two-day CPET studies) [23]. However, considering the possible differences in the results of one-day and two-day CPET, two-day CPET data may need to be analyzed separately in various parameters.

Therefore, we meta-analyzed the parameters of VO_2_ and workload both at peak (VO_2peak_, Workload_peak_) and at ventilatory threshold (VO_2_@VT, Workload@VT) in two-day CPET studies to assess the potential use of this test as a diagnostic tool for ME/CFS. 

## 2. Methods 

### 2.1. Search Strategy, Inclusion and Exclusion Criteria

We conducted a systematic search in five public databases and two search sources: PubMed, the Cochrane Library, the Cumulative Index to Nursing and Allied Health Literature (CINAHL), Medline, Google Scholar, a hand search, and the reference lists of the included studies. The search was conducted from February to June 2020. The search keywords were ‘Chronic fatigue syndrome’ [MeSH terms] AND ‘cardiopulmonary exercise test’ or ‘CEPT’ [MeSH terms]. For the Google Scholar search in particular, we restricted the search criteria to reduce the initial abundant numbers and selected the studies not searched by PubMed. Papers were screened using the following inclusion criteria: (a) studies of cross-sectional, case–control, prospective studies that conducted the CPET to measure diagnostic parameters for ME/CFS, (b) two-day CPET with the two sessions conducted 24 h apart, (c) studies that measured the four parameters (VO_2peak_, Workload_peak_, VO_2_@VT, and Workload@VT) for the test. The exclusion criteria were as follows: (a) non clinical-based studies, (b) studies for other than ME/CFS, and studies with (c) single CPET, (d) no controls and that failed to state the four parameter values, (e) less than 5 participants, (f) studies that measured other than the four parameters. The search and data extraction were performed by E-J. L. and C-G. S., and any disagreements were resolved by discussions. 

### 2.2. Data Extraction and Outcome Measures

The main features of the studies were extracted and compiled including; authors, year of publication, and the number of participants, the age and body mass index (BMI) of the participants, and the methodology of selecting the participants (Table 1). The primary outcome measures were the four parameters: VO_2peak_, Workload_peak_, VO_2_@VT, and Workload@VT. These parameters are involved in measuring activity limitations of ME/CFS patients [24]. These are defined as follows: (1) VO_2peak_ is the highest value of oxygen uptake obtained during the exercise. (2) VO_2_ at ventilatory threshold (VT) is the volume of oxygen at VT, which is the point which ventilation starts to increase at a faster rate than oxygen consumption. (3) Workload_peak_ is the power output produced by the participant at peak. (4) Workload@VT is the level of power output produced at VT [11,14]. 

### 2.3. Statistical Analysis 

We calculated the mean difference and meta-analyzed the four parameters of ME/CFS patients and controls as of Testes 1 and 2 (Table 2, Table 3 and Table 4). The four parameters was estimated with a random-effects model to account for the heterogeneity of the data, and we compared the *I*^2^ and significance (*p* values) of those groups. *I*^2^ higher than 50% is considered substantially heterogeneous [25]. A forest plot was used to show the estimated overall mean number of the parameters, and *p* value < 0.05 is considered significant. The data were analyzed using the meta-analysis program version 5.3 Review Manager (The Cochrane Collaboration, Oxford, UK) [26].

### 2.4. Assessment of Quality of Studies 

The quality of studies included was assessed according to the Newcastle Ottawa Scale (NOS) guideline for case-control studies, which judges studies on the following perspectives: selection of the groups, the comparability of the groups, and the ascertainment of exposure [27] (Table 1).

## 3. Results

### 3.1. Characteristics of Included Studies 

In total, 218 studies were initially selected as candidates for this meta-analysis; ultimately, 5 studies were included (Figure 1). All studies used the same study protocol (an incremental cycling protocol with an interval of 24 h) and case definition (Fukuda) for inclusion of participants. Two studies additionally used the International Consensus Criteria (ICC) [28] and/or the Canadian Consensus Criteria (CCC) [6]. The total number of participants across the five studies was 98 ME/CFS patients (90 female and 8 male) and 51 controls (45 female and 6 male). The total mean age and BMI of the participants were comparable between the patient and control groups (Table 1).

### 3.2. Comparisons of the Four Parameters 

The four parameters as of Test 1 and Test 2, as well as their differences between sessions, are summarized in Table 2. From Test 1 to Test 2, all four measures (VO_2peak_, VO_2_@VT, Workload_peak_, and Workload@VT) increased in the control group but decreased in the patient group. Among the four parameters, the Workload@VT of the patient group showed an especially marked decrease on Test 2 compared to Test 1, with a difference (Test 2–Test 1) of −14.6 in patients and +6.5 in controls (Table 2).

### 3.3. Outcomes of Meta-Analysis

From the meta-analysis, we evaluated the differences in parameters from Test 2 to Test 1 in ME/CFS patients compared to the control group (Figure 2). The values of all four parameters increased as of the 2nd test (Test 2) in the control group, while ME/CFS patients showed notable decreases in all parameters at Test 2 (Figure 2, Table 2 and Table 3). 

In general, the differences between patients and controls were greater at VT than at peak and greater for workload than for VO_2_ (Figure 2 and Figure 3). From the meta-analysis focused on the difference between Test 2–Test 1 (using the data from patients and controls), Workload@VT showed the most notable significant difference (*p* < 0.05) (Figure 2 and Figure 3). 

## 4. Discussion

From our meta-analysis of five studies of two-day CPET, we identified the following three key features. First, ME/CFS patients appeared to have lower levels of all parameters than controls, especially on Test 2. Second, on Test 2, the difference between the patients and controls was observed to be larger at VT than at peak. Third, on a more specific level, Workload@VT was notably different between the two tests and between the two groups (Figure 2 and Figure 3, Table 2, Table 3 and Table 4). These data may indicate the potential of Workload@VT value as an objective measurement in ME/CFS. 

The reduced levels of parameters in the ME/CFS patient group on Test 2 shows that they failed to reproduce their work capacity from Test 1 [19]. Such reduced work capacity in general is likely linked to a lack of cellular adenosine triphosphate (ATP) production, which normally occurs through aerobic and anaerobic metabolism [31]. Imbalanced ATP production can be caused by functional impairment of mitochondria in patients with ME/CFS [32]. Studies have reported conflicted results regarding mitochondrial impairment; some have provided evidence of actual mitochondrial dysfunction affecting oxidative phosphorylation [21,33], while other studies found no abnormal indicators of altered mitochondrial function or mitochondrial DNA mutation [19,34,35]. In fact, a number of researchers suggested a possible problem in the pathway of oxygen or glucose transportation into the cells, which may inhibit the function of mitochondria in ME/CFS patients [10,19,29,30,35,36]. The hypothesized factors inhibiting mitochondria ATP production include viral/bacterial infection [37], high levels of proinflammatory cytokines [38], reactive oxygen species (ROS) [39], and decreased levels of enzymes (such as pyruvate dehydrogenase, PDH) needed in the process of aerobic cellular respiration [40].

Lower work capacity on Test 2 than on Test 1 seems to be a unique feature of ME/CFS and is indicated as a cardinal feature for the diagnosis of PEM [3]. Patients with lung, heart, or kidney diseases presented no significant differences between Tests 1 and 2 in repeated CPETs [41,42,43]. Moreover, patients with other fatigue-inducing conditions such as multiple sclerosis (MS) and positive human immunodeficiency virus (HIV) status showed improved Workload@VT values on Test 2, in contrast to ME/CFS patients [30,44]. The present meta-analysis consists exclusively of data from studies containing control groups. Several other studies, despite having no control groups with larger patient group [45,46], and male ME/CFS patient group [46], produced similar outcomes, with ME/CFS patients’ lower values of Workload@VT on Test 2 than on Test 1. 

Although mitochondria-associated alterations have been observed, the underlying pathophysiology of PEM has not been explored in detail. Most recently, several groups have investigated metabolic changes in ME/CFS patients with PEM [12,47]. Naviaux et al. found reduced concentrations of specific metabolites in the plasma of 45 ME/CFS patients [47]. McGregor et al. also found altered glycolysis, a low level of acetate, and a positive correlation between urine metabolites and the severity of PEM [12]. These metabolite-based findings have been applied to the development of diagnostic tools. The metabolic response showed high validity in a diagnostic test for ME/CFS, achieving accuracy of over 95% [47]. Esfandyarpour et al. demonstrated the use of a nanoneedle to measure a metabolite-based biomarker from a small volume of blood in ME/CFS patients [48]. The present data might support a link between PEM symptom and alterations in metabolism and mitochondrial ATP production in ME/CFS. 

## 5. Conclusions

The meta-analysis indicates a significant alteration of workload at VT especially on the 2nd day of CPET in ME/CFS patients. Accordingly, the two-day CPET could be considered as one of the potential objective assessment tools for PEM in ME/CFS patients. The present study may be of value in suggesting a direction for the development of ME/CFS diagnostic tools. However, we should consider the limitations of this study, the relatively small number of participants and the studies included. Additionally, the absence of percent predicted values for different genders should also be noted as another weak point of this study. For further studies, the following should be cautiously considered: the selection of the participants using appropriate diagnostic criteria, the severity of symptoms, the comparison of males and females, and the comparison with other fatigue-inducing disorders. Studies at larger scales with more rigorous methodology are needed. 

## Figures and Tables

**Figure 1 jcm-09-04040-f001:**
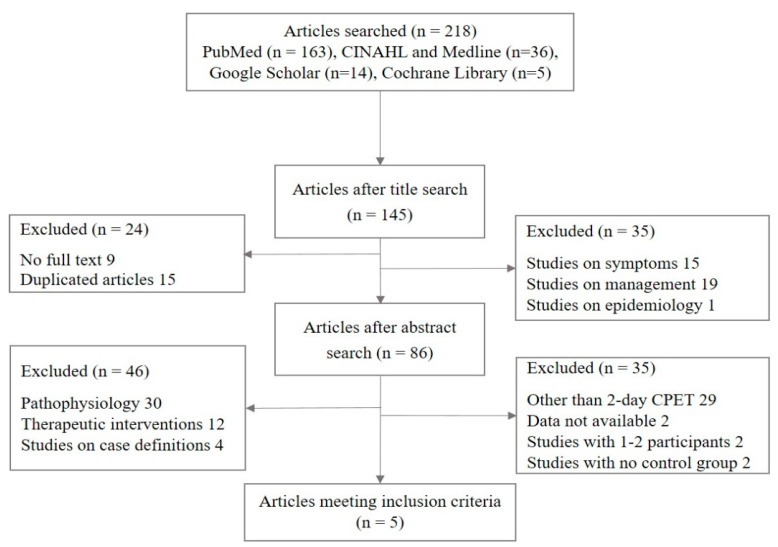
Flow chart of the study selection process.

**Figure 2 jcm-09-04040-f002:**
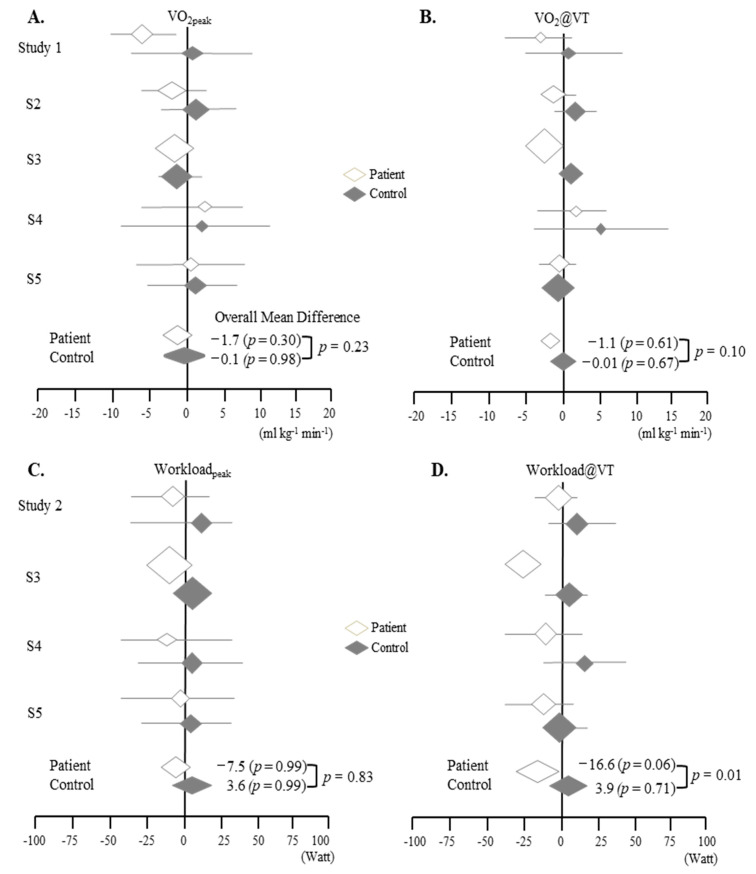
Meta analyzed overall mean differences of the four parameters (A-D) from the scores of the two-day CPET (test 2–test 1) for the ME/CFS (Myalgic encephalomyelitis/chronic fatigue syndrome) patients and controls: (**A**). VO_2_peak, (**B**). VO_2_@VT, (**C**). Workloadpeak, (**D**). Workload@VT. *p*, *p* value. Refer to Table 3 for details.

**Figure 3 jcm-09-04040-f003:**
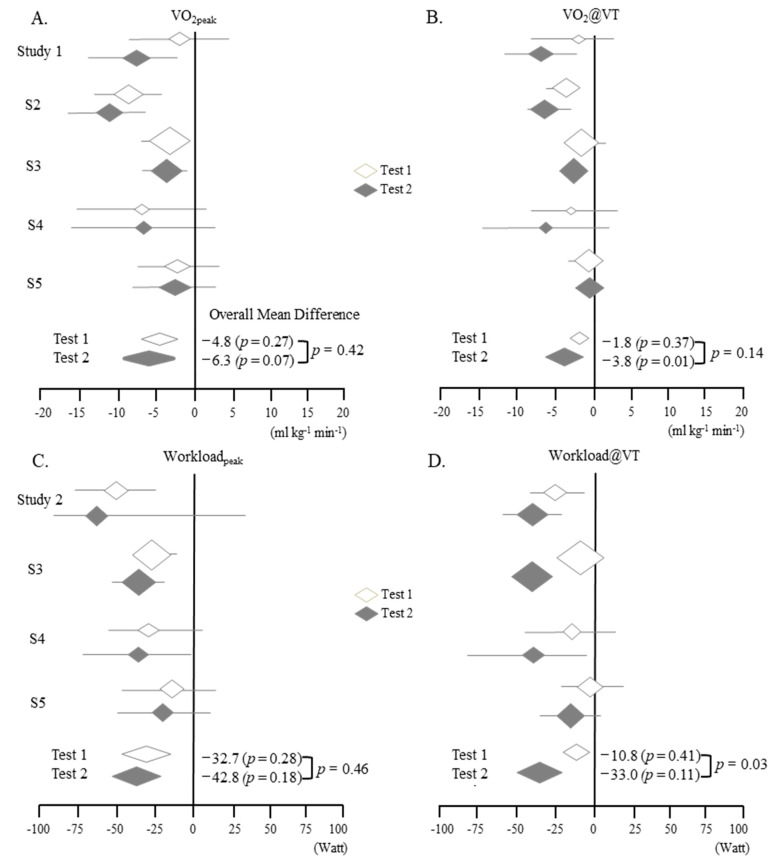
Meta analyzed overall mean differences of the four parameters (A-D) from the scores of the ME/CFS patients―controls for the two-day CPET 1 and 2: (**A**). VO_2_peak, (**B**). VO_2_@VT, (**C**). Workloadpeak, (**D**). Workload@VT. *p*, *p* value. Refer to Table 4 for details.

**Table 1 jcm-09-04040-t001:** Characteristics and assessment of quality of studies.

	ME/CFS +	Control	Newcastle Ottawa Scale ^§^							
			S				C		E	
			1	2	3	4	5	6	7	8
No. of studies	5	5								
No. of participants (Female/Male)	98 (90/8)	51 (45/6)								
Age (y) ^#^	42.3 ± 11.6	41.3 ± 12.4								
BMI (kg/m^2^) ^#^	25.3 ± 4.8	25.6 ± 4.2								
	Selection of participants									
	ME/CFS	Control								
Van Ness (2007) [10]	Physician diagnosis and Fukuda	Healthy sedentary	*	*	*	*	*	*	*	*
Vermeulen (2010) [19]	CDC-SI 59.5 ± 13.1 ^‡^ Fukuda, infectious disease onset	CDC-SI 5.0 ± 4.5	*	*	*	*	*	*	*	*
Snell (2013) [29]	Sedentary ^※^ Fukuda, Presence of PEM	Sedentary ^※^	*	*	*	*	*	*	*	*
Hodges (2017) [30]	Fukuda, (CCC and ICC) ^α^ De Paul Questionnaire, SF-36	Healthy	*	*	-	-	*	*	*	*
Nelson (2019) [20]	Sedentary ^†^ Physician diagnosis Fukuda/(CCC/ICC) ^α^	Sedentary ^†^	*	*	*	*	*	*	*	*

+ ME/CFS, Myalgic encephalomyelitis/chronic fatigue syndrome. ^§^ A star system at the included studies are judged on 3 perspectives: S, selection of groups. C, comparability of groups. * checked and confirmed. E, exposure. CDC-SI, Centers for Disease Control and Prevention-Symptom Inventory questionnaire. ^#^ Estimated from the mean ages of the individual studies. ^‡^ Four ME/CFS fatigue symptoms ≥7.5. ^※^ <30 min exercise/per week. ^†^ <150 min of moderate exercise/per week. ^α^ CCC, Canadian consensus criteria. ICC, International consensus criteria.

**Table 2 jcm-09-04040-t002:** Studies that performed the two-day cardiopulmonary exercise test (CPET) in ME/CFS patients and controls.

#	Study	Participants (Patients/ Controls)	VO_2peak_	VO_2_@VT	Workload_peak_	Workload@VT
			Test 1/Test 2 (T2-T1)	Test 1/Test 2 (T2-T1)	Test 1/Test 2 (T2-T1)	Test 1/Test 2 (T2-T1)
			(mL kg^−1^ min^−1^)	(mL kg^−1^ min^−1^)	(Watt)	(Watt)
1	Van Ness (2007) [10]	6(F) 6(F)	26.2 ± 4.9/20.5 ± 1.8 (−5.8) 28.4 ± 7.2/28.9 ± 8.0 (+0.5)	15.0 ± 4.9/11.0 ± 3.4 (−4.0) 17.6 ± 4.9/18.0 ± 5.3 (+0.4)	- -	- -
2	Vermeulen (2010) [19]	15(F) 15(F)	22.3 ± 5.7/20.9 ± 5.5 (−1.3) 31.2 ± 7.0/31.9 ± 7.4 (+0.7)	12.8 ± 3.0/11.9 ± 2.9 (−0.9) 16.7 ± 4.0/18.0 ± 4.6 (+1.3)	132.0 ± 30.0/125.0 ± 35.0 (−7.0) 188.0 ± 46.0/196.0 ± 51.0 (+8.0)	58.6 ± 24.2/54.5 ± 20.9 (−4.1) 82.9 ± 29.1/92.9 ± 31.4 (+10.0)
3	Snell (2013) [29]	51(F) 10(F)	21.5 ± 4.1/20.4 ± 4.5 (−1.1) 25.0 ± 4.4/24.0 ± 4.3 (−1.0)	12.7 ± 2.9/11.4 ± 2.9 (−1.3) 13.8 ± 2.8/14.1 ± 3.3 (+0.3)	109.6 ± 28.9/101.6 ± 30.7 (−8.0) 137.2 ± 23.2/140.0 ± 25.0 (+2.8)	49.5 ± 20.4/22.2 ± 18.1 (−27.3) 58.0 ± 16.7/63.5 ± 19.5 (+5.5)
4	Hodges (2017) [30]	9(F)/1(M) 9(F)/1(M)	25.0 ± 8.9/26.3 ± 7.8 (+1.3) 32.0 ± 10.9/33.1 ± 12.5 (+1.1)	21.0 ± 4.3/22.2 ± 6.2 (+1.2) 23.6 ± 9.0/28.5 ± 12.5 (+4.9)	135.0 ± 43.0/126.0 ± 45.0 (−9.0) 164.0 ± 40.0/167.0 ± 41.0 (+3.0)	105.0 ± 30.0/93.0 ± 37.0 (−12.0) 119.0 ± 28.0/132.0 ± 42.0 (+13.0)
5	Nelson (2019) [20]	9(F)/7(M) 5(F)/5(M)	27.3 ± 9.2/27.4 ± 8.8 (+0.1) 29.9 ± 6.1/30.3 ± 6.2 (+0.4)	15.9 ± 4.1/15.4 ± 3.4 (−0.5) 16.5 ± 2.0/15.9 ± 1.5 (−0.6)	154.4 ± 56.0/152.5 ± 51.7 (−1.9) 172.0 ± 35.5/174.0 ± 36.6 (+2.0)	87.8 ± 29.6/72.5 ± 27.7 (−15.3) 90.5 ± 17.1/88.0 ± 16.7 (−2.5)
Total 90(F)/8(M) 45(F)/6(M)			24.5 ± 6.6/23.1 ± 5.7 (−1.4) 29.3 ± 7.1/29.6 ± 7.7 (+0.3)	15.5 ± 3.8/14.4 ± 3.8 (−1.1) 17.6 ± 4.5/18.9 ± 5.4 (+1.3)	132.8 ± 39.5/126.3 ± 40.6 (−6.5) 165.3 ± 36.2/169.3 ± 38.4 (+4.0)	75.2 ± 26.1/60.6 ± 25.9 (−14.6) 87.6 ± 22.7/94.1 ± 27.4 (+6.5)

ME/CFS, Myalgic encephalomyelitis/chronic fatigue syndrome. VO_2peak_, volume of oxygen uptake at peak. @VT, at ventilatory threshold. The units of parameter for VO_2_ is mL kg^−1^ min^−1^, and for Workload is Watt. F, female. M, male. All results are rounded to one decimal place. Exercise mode: cycle ergometer.

**Table 3 jcm-09-04040-t003:** Meta-analysis of two-day CPET (test 2–test 1) for the comparisons between the ME/CFS patients and controls.

#	Study		Weight (%)				Mean Difference			
			VO_2peak_	@VT	WL_peak_	@VT	VO_2peak_	@VT	WL_peak_	@VT
1	Van Ness (2007) [10]	P	16	3.5	-	-	−5.7 (−1.5, −9.9)	−4.0 (0.8, −8.8)	-	-
		C	8.2	4.3	-	-	0.5 (9.1, −8.1)	0.4 (6.2, −5.4)	-	-
2	Vermeulen (2010) [19]	P	17.1	17.9	17.2	26.2	−0.9 (1.2, −3.0)	−1.4 (2.6, −5.4)	−7.0 (16.3, −30.3)	−4.1 (12.1, −20.3)
		C	23.0	15.0	17.0	18.4	0.7 (5.9, −4.5)	1.3 (4.4, −1.8)	8.0 (42.8, −26.8)	10.0 (31.6, −11.6)
3	Snell (2013) [29]	P	53.1	63.1	69.8	38.6	−1.1 (0.6, −2.8)	−1.3 ( −0.2, −2.4)	−8.0 (3.6, −19.6)	−27.3 (−19.8, −34.8)
		C	42.0	19.8	46.0	33.8	−1.0 (2.8, −4.8)	0.3 (3.0, −2.4)	2.8 (23.9, −18.3)	5.5 (21.4, −10.4)
4	Hodges (2017) [30]	P	5.9	3.7	6.3	13.4	1.3 (8.6, −6.0)	1.2 (5.9, −3.5)	−9.0 (29.6, −47.6)	−12.0 (17.5, −41.5)
		C	5.8	1.6	16.3	8.7	1.1 (11.4, −9.2)	4.9 (14.5, −4.7)	3.0 (38.5, −32.5)	13.0 (44.3, −18.3)
5	Nelson (2019) [20]	P	7.9	11.7	6.7	21.7	0.1 (6.3, −6.1)	−0.5 (2.1, −3.1)	−1.9 (35.5, −39.3)	−15.3 (4.6, −35.2)
		C	21.0	59.4	20.6	39.0	0.4 (5.8, −5.0)	−0.6 (1.0, −2.2)	2.0 (33.6, −29.6)	−2.5 (12.3, −17.3)
Overall values for P Heterogeneity (*I*^2^) *p* value			19% 0.30	0 0.61	0% 0.99	60% 0.06	−1.7 (0.2, −3.5)	−1.1 (−0.2, −2.0)	−7.5 (2.2, −17.2)	−16.6 (−3.6, −29.5)
Overall values for C Heterogeneity (*I*^2^) *p* value			0% 0.98	0% 0.67	0% 0.99	0% 0.71	−0.1 (2.4, −2.5)	−0.01 (1.2, −1.2)	3.6 (17.9, −10.8)	3.9 (13.1, −5.4)
P vs. C *p* value			0.23	0.10	0.83	0.01 **				

ME/CFS, Myalgic encephalomyelitis/chronic fatigue syndrome. CPET, Cardiopulmonary exercise testing. #, Number of studies. VO_2peak_, volume of oxygen uptake at peak. @VT, at ventilatory threshold. WL _peak,_ workload at peak. The units of parameter for VO_2_ is mL × kg^−1^ × min^−1^, and for Workload or WL is Watt. P, patient. C, control. All results are rounded to one decimal place. **, *p* value < 0.01.

**Table 4 jcm-09-04040-t004:** Meta-analysis (of ME/CFS patient control) for the comparisons between the two-day CPET 1 and 2.

#	Study		Weight (%)				Mean Difference			
			VO_2peak_	@VT	WL_peak_	@VT	VO_2peak_	@VT	WL_peak_	@VT
1	VanNess (2007) [19]	T1	11.9	5.4	-	-	−2.2 (−9.2, 4.8)	−2.6 (−8.1, 2.9)	-	-
		T2	16.4	15	-	-	−8.4 (−14.7, −1.8)	−7.0 (−12.0, −2.0)	-	-
2	Vermeulen (2010) [19]	T1	23.4	23.9	22.8	18.7	−8.9 (−13.37, −4.3)	−3.9 (−6.4, −1.4)	−56.0 (−83.79, −28.2)	−24.3 (−43.5, −5.2)
		T2	23.1	23.9	22.3	25.5	−11.0 (−15.7, −6.3)	−6.1 (−8.9, −3.4)	−71.0 (−102.3, 39.7)	−38.4 (−57.4, −19.4)
3	Snell (2013) [29]	T1	41.0	39.3	47.2	49.5	−3.5 (−6.5, −0.6)	−1.1 (−3.0, 0.8)	−27.6 (−44.0, −11.2)	−8.5 (−20.3, 3.3)
		T2	31.0	26.3	40.5	34.8	−3.6 (−6.5, −0.7)	−2.7 (−4.9, −0.5)	−38.4 (−56.0, −20.8)	−41.3 (−54.4, −28.2)
4	Hodges (2017) [30]	T1	8.0	4.3	14.6	10.6	−7.0 (−15.7, 1.7)	−2.6 (−8.8, 3.6)	−29.0 (−65.4, 7.4)	−14.0 (−39.4, 11.4)
		T2	10.5	7.4	17.2	11.6	−6.8 (−15.9, 2.3)	−6.3 (−15.0, 2.4)	−41.0 (−78.7, −3.3)	−39.0 (−73.7, −4.3)
5	Nelson (2019) [20]	T1	15.8	27.1	15.4	21.2	−2.6 (−8.5, 3.3)	−0.6 (−3.0, 1.8)	−17.6 (−52.8, 17.6)	−2.7 (−20.7, 15.3)
		T2	18.9	27.4	20.0	28.2	−2.9 (−8.7, 2.9)	−0.5 (−2.4, 1.4)	−21.5 (−55.5, 12.5)	−15.5 (−32.6, 1.6)
Overall values for T1 Heterogeneity (*I*^2^) T1 *p* value			22% 0.27	7% 0.37	21% 0.28	0% 0.41	−4.8 (−7.3, −2.2)	−1.8 (−3.1, −0.5)	−3–32.7 (−47.6, −17.8)	−10.8 (−19.1, −2.5)
Overall values for T2 Heterogeneity (*I*^2^) T2 *p* value			53% 0.07	72% 0.01**	38% 0.18	50% 0.11	−6.3 (−9.8, −2.9)	−3.8 (−6.5, −1.1)	−42.8 (−61.0, −24.5)	−33.0 (−46.4, −19.6)
T1 vs. T2 *p* value			0.42	0.14	0.46	0.03 *				

ME/CFS, Myalgic encephalomyelitis/chronic fatigue syndrome. CPET, Cardiopulmonary exercise testing. VO_2peak_, volume of oxygen uptake at peak. @VT, at ventilatory threshold. WL _peak,_ workload at peak. The units of parameter for VO_2_ is mL kg^−1^ min^−1^, and for Workload or WL is Watt. T1, Test 1. T2, Test 2. All results are rounded to one decimal place. *, *p* value < 0.05. **, *p* value < 0.01.

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
