# Peer review of "The Prospects of the Two-Day Cardiopulmonary Exercise Test (CPET) in ME/CFS Patients: A Meta-Analysis"

_jcm, 2020, doi:10.3390/jcm9124040_

Round 1

Reviewer 1 Report

As male and females have different values for oxygen consumption mentioned and discussed in several articles, comparing absolute oxygen consumption in studies with both male and female controls and patients is not done.

When pointing out studies without controls and not referring to the study with the largest population of 2-day cpet is an ommission.

abnormalities on day 2 in only worklaod VT is probably in not a very large population not enough to conclude the potential for post-exertional malaise detection.

The main topic of discussion is atp/mitochondrial abnormalities. As many of the features of ME/CFS is multifactorial, PEM probably is too. 

Author Response

Response to Reviewer 1 Comments

1. As male and females have different values for oxygen consumption mentioned and discussed in several articles, comparing absolute oxygen consumption in studies with both male and female controls and patients is not done.

Response 1: Thank reviewer for the comment. We collected and included studies which compared ME/CFS patients and controls for analysis of the four parameters. We focused on the differences between CPET1 and 2 of those groups. Also, the majority of the participants in this study (91% in patient and 87% in control group) are female. According to van Campen (2020)’s study, the reduction between the tests was observed for male ME/CFS patients likely female. We clarified this in the Discussion on page 10.

2. When pointing out studies without controls and not referring to the study with the largest population of 2-day cpet is an omission.

Response 2: Thank reviewer for the detailed review. As following reviewer’s suggestion, we have amended the statement (~Several other studies, despite having no control groups, larger patient group, and male ME/CFS patient group ~) in the Discussion section on page 10.

3. Abnormalities on day 2 in only worklaod VT is probably in not a very large population not enough to conclude the potential for post-exertional malaise detection.

Response 3: Thank reviewer for the professional comment. We have amended and clarified the limitations of this study in the Conclusion section on page 10. This study is limited to the small number of participants and the studies included. In this study, we aimed to highlight the significant difference of the parameter (reduced work capacity) for the ME/CFS patients on the 2nd test, and suggest consider the potential use of the test to discriminate ME/CFS patients from normal population. Further studies in larger scales should be conducted with various other parameters.

4. The main topic of discussion is atp/mitochondrial abnormalities. As many of the features of ME/CFS is multifactorial, PEM probably is too.

Response 4: Thank reviewer for the professional comment. The pathophysiology of PEM has not been established, as reviewer pointed out. We have described this in the present revised manuscript on page 10.

Reviewer 2 Report

Background

Well written but additional explanation of the chosen parameters for analysis within the MA is warranted – why have a mixture of peak and VT parameters been selected in particular and why have these been investigated in prior studies.

Method

The method is missing numerous important details, I suggest the following additions and clarifications to the authors:

Was the study conducted according to PRISMA (Preferred reporting items for systematic reviews and meta-analyses) guidelines? If not, a statement of what aspects of these guidelines were not adhered to may be warranted

Please provide the date of search

Please provide additional details about the conduct of the search – were studies screened by one or two reviewers, what software or otherwise was utilised to perform the screening (e.g. covidence), if multiple reviewers were utilised how were conflicts between reviewers settled, etc

There is a need for additional information within the exclusion/inclusion criteria – was there a particular age cut-off for participants that was utilised to determine if a study was eligible? Were particular diagnostic criteria accepted and others not? If no reputable diagnostic criteria was reported in a study, was it still deemed to be potentially eligible?

I think you should refrain from saying that workload is the ‘energy released’ (line 80) as this is an erroneous statement in bioenergetic terms. A more appropriate definition would be the ‘power output produced by the participant’ at the various timepoints. Additionally, this is not necessarily calculated using the formula of resistance, distance and rpm depending on the ergometer used and the method it uses to apply and measure resistance/force, so can remove the statement to this effect in line 81.

Were any mean difference or standardised mean difference calculated as part of the analysis? I assume so in order for a p-value to have been calculated? The way your method is current written suggests that this wasn’t done as part of the analysis (line 86, suggests only the heterogeneity and p values were looked at). Looking at the results it appears that MD was calculated but this must be stated in the method.

Please provide further details about RevMan program as appropriate (e.g. company, location etc).

Results

What do you mean you ‘selected’ 218 studies (line 91)? You should be cleared about how many original records were generated at all stages (e.g. from initial search, after duplicates were screened, during title and abstract screening, during full-text screening etc. If this is included in the supplementary section, this should be more clearly stated within the context of where these results are reported, not just at the end of the overall results section .See Nelson, et al., 2019. Evidence of altered cardiac autonomic regulation in myalgic encephalomyelitis/chronic fatigue syndrome: A systematic review and meta-analysis. Medicine, 98(43) for an example of correct reporting of these details in an article from the same population. You should report the number of total results - I would imagine from google scholar alone would have given you thousands of results – how did you deal with these?

Given you extracted information on the age, n, bmi, diagnostic criteria etc, this should be reported as part of the method. Currently you only report that data on the VO2/workload parameters

Table 2 is completely devoid of units – this makes it impossible to interpret the data – must be rectified

In my opinion, meta-analysis results should be changed so that the change seen in the patient group is shown to be a DECREASE on the forest plots on day 2, rather than them showing an increased score on day one. This would be more intuitive.

Discussion

The authors have done a good job of providing an overview of the findings and potential underlying physiology.

As with the method, avoid saying ‘energy production’ (Line 152) and focus specifically on it being a lower capacity for work.

Please provide further insight into the strengths and weaknesses of this analysis and potential cautionary points in the interpretation, particularly based on a relatively low sample for an analysis of this type

Please include a clearer conclusion to summarise the main findings and potential practical implications of this work as this is not currently apparent at the end of the paper.

Author Response

Response to Reviewer 2 Comments

1. Well written but additional explanation of the chosen parameters for analysis within the MA is warranted – why have a mixture of peak and VT parameters been selected in particular and why have these been investigated in prior studies.

Response 1: Thank reviewer for the professional comment and questions. We have added an explanation to justify the analysis of the parameters in the Method section 2.2 on page 2. The parameters of VO2 and workload both at peak (VO2peak, Workloadpeak) and at ventilatory threshold (VO2@VT, Workload@VT) in two-day CPET are an important measure for understanding the activity limitations in ME/CFS. In this study, we aimed to measure the work capacity of ME/CFS in comparison to the control group.

2. Was the study conducted according to PRISMA (Preferred reporting items for systematic reviews and meta-analyses) guidelines? If not, a statement of what aspects of these guidelines were not adhered to may be warranted. Please provide the date of search

Response 2: Thank reviewer for the helpful suggestion. I have moved the supplementary PRISMA flow chart to the main text as Figure 1 on page 3. Also, the date of search has been added in the Method section 2.1 on page 2.

3. Please provide additional details about the conduct of the search – were studies screened by one or two reviewers, what software (e.g. covidence), how were conflicts between reviewers settled, etc

Response 3: Thank reviewer for the helpful suggestion. I have amended and added the details for search methodology in the Methods 2.1 on page 2.

4. There is a need for additional information within the exclusion/inclusion criteria – was there a particular age cut-off for participants that was utilised to determine if a study was eligible? Were particular diagnostic criteria accepted and others not? If no reputable diagnostic criteria was reported in a study, was it still deemed to be potentially eligible?

Response 4: As stated above response 2, the exclusion and inclusion criteria are restated in more detail in Method section on page 2~3. No restrictions on the age of the participants and diagnostic criteria were applied for this study.

5. I think you should refrain from saying that workload is the ‘energy released’ (line 80) as this is an erroneous statement in bioenergetic terms. A more appropriate definition would be the ‘power output produced by the participant’ at the various time points. Additionally, this is not necessarily calculated using the formula of resistance, distance and rpm depending on the ergometer used and the method it uses to apply and measure resistance/force, so can remove the statement to this effect in line 81.

Response 5: We really appreciate reviewer for the professional comments. We have removed the parts you have mentioned (line 80 ~81), and added the definition you suggested.

6. Were any mean difference or standardized mean difference calculated as part of the analysis? I assume so in order for a p-value to have been calculated? The way your method is current written suggests that this wasn’t done as part of the analysis (line 86, suggests only the heterogeneity and p values were looked at). Looking at the results it appears that MD was calculated but this must be stated in the method.

Response 6: Thank reviewer for the kind suggestion. We added the additional information in ‘Method’ section 2.3, and have moved the initially supplementary Table 1 and 2 to the main text as Table 3 and 4, on page 7 and 9.

7. Please provide further details about RevMan program as appropriate (e.g. company, location etc).

Response 7: We have added the detailed information for the program on page 3.

8. What do you mean you ‘selected’ 218 studies (line 91)? You should be cleared about how many original records were generated at all stages (e.g. from initial search, after duplicates were screened, during title and abstract screening, during full-text screening etc. If this is included in the supplementary section, this should be more clearly stated within the context of where these results are reported, not just at the end of the overall results section. See Nelson, et al., 2019. Evidence of altered cardiac autonomic regulation in myalgic encephalomyelitis/chronic fatigue syndrome: A systematic review and meta-analysis. Medicine, 98(43) for an example of correct reporting of these details in an article from the same population. You should report the number of total results - I would imagine from google scholar alone would have given you thousands of results – how did you deal with these?

Response 8: Thank reviewer for the helpful comments. As reviewer requested, we added the detailed description of selection of studies in ‘Method’ section 2.1 on page 2. Especially for Google Scholar search, we restricted the search criteria to reduce the initial abundant numbers and selected only studies not searched by PubMed.

9. Given you extracted information on the age, n, bmi, diagnostic criteria etc, this should be reported as part of the method. Currently you only report that data on the VO2/workload parameters

Response 9: We have added a statement to explain the characteristics of the studies shown in Table 1, in 2.2 Methods, on page 2.

10. Table 2 is completely devoid of units – this makes it impossible to interpret the data – must be rectified

Response 10: Thank reviewer for your detailed review. I have added the units of the parameters in the Table 2.

11. In my opinion, meta-analysis results should be changed so that the change seen in the patient group is shown to be a DECREASE on the forest plots on day 2, rather than them showing an increased score on day one. This would be more intuitive.

Response 11: Thank reviewer for the helpful comment. We carefully guess that reviewer might have misunderstood the indications of Figure 2 and 3 (forest plots). We described the differences of CPET 1 and 2 between the ME/CFS patients and controls (Figure 2) and vice versa for Figure 3 (the differences of patients and controls between CPET1 and 2). Thus, the larger scores in the forest plots indicate the bigger differences between the CPET 1 and 2 for the ME/CFS patients (Figure 2). I clarified this in the legends of the figure 2 and 3, also in the context on page 5 and 7.

12. The authors have done a good job of providing an overview of the findings and potential underlying physiology. As with the method, avoid saying ‘energy production’ (Line 152) and focus specifically on it being a lower capacity for work. Please provide further insight into the strengths and weaknesses of this analysis and potential cautionary points in the interpretation, particularly based on a relatively low sample for an analysis of this type

Response 12: Thank you for your kind suggestions. As you have suggested, I have changed the wordings for ‘energy production’ to ‘lower work capacity’ on page 10. I also have made amendment for the Conclusions, and clarified the strengths and weakness, limitations of this study on page 10.  

13. Please include a clearer conclusion to summarize the main findings and

potential practical implications of this work as this is not currently apparent at the end of the paper.

Response 13: Thank reviewer for your thorough review. We have summarized the main findings and clarified the Conclusion of this study on page 10.

Reviewer 3 Report

The topic of this manuscript may be of interest to researchers, clinicians, and patients in the ME/CFS community, however, this reviewer questions the need for a meta-analysis when so few studies have been conducted that meet the authors’ eligibility criteria. Five studies is a very small number for such a review and also does not allow for the proper conduct of a meta-regression analysis which often requires 10 or more studies for readers to draw meaningful conclusions.

Guidelines for proper conduct and reporting of meta-analysis are available and should be followed http://www.prisma-statement.org/PRISMAStatement/ . I Strongly suggest the authors review this document and make their manuscript as consistent as possible with these guidelines.

The reporting of the methods is extremely minimal, making replication and comparability of this study difficult. Several things should be included, such as (i) inclusion and exclusion criteria for selecting articles should be more explicitly stated, (ii) how the effects were weighted, (iii) which type of effect size was selected and why, (iv) assessment of publication bias, (v) assessment of study quality, and many others.

Author Response

Response to Reviewer 3 Comments

1. The topic of this manuscript may be of interest to researchers, clinicians, and patients in the ME/CFS community, however, this reviewer questions the need for a meta-analysis when so few studies have been conducted that meet the authors’ eligibility criteria. Five studies is a very small number for such a review and also does not allow for the proper conduct of a meta-regression analysis which often requires 10 or more studies for readers to draw meaningful conclusions.

Response 1: Thank reviewer for the positive and critical comments. As reviewer indicated, the small number of studies is a main limitation in present study. We clarified this limitation in the text.

2. Guidelines for proper conduct and reporting of meta-analysis are available and should be followed http://www.prisma-statement.org/PRISMAStatement/. I Strongly suggest the authors review this document and make their manuscript as consistent as possible with these guidelines.

Response 2: Thank you for the suggestion. I have moved the initial supplementary figure of flow chart of the study selection process to the main text on page 3 (Figure 1). The Methods sections are also amended and added more information in detail (inclusion and exclusion criteria, data extraction, and analysis, etc…), on page 2~3.

3. The reporting of the methods is extremely minimal, making replication and comparability of this study difficult. Several things should be included, such as (i) inclusion and exclusion criteria for selecting articles should be more explicitly stated, (ii) how the effects were weighted, (iii) which type of effect size was selected and why, (iv) assessment of publication bias, (v) assessment of study quality, and many others.

Response 3: Thank reviewer for the detailed comments. In present revised manuscript, we have amended the ‘Methods’ section. Please refer to page 2~3, Figure 1, and Table 1.

Round 2

Reviewer 1 Report

I took a look at the updated version and the comments I had on the first version are still valid: with the time to update the largest 2 day study (without controls) should have been referred to. The data from the included studies are small and any comments on absolute oxygen consumption values in a study combining male and female patients (without having information on percent predicted values) are scientifically not sound.

Author Response

  1. With the time to update the largest 2 day study (without controls) should have been referred to.

Response 1: Thank you for the comment. As you suggested, I have added the references for the studies with larger group without controls in the text.

  1. The data from the included studies are small and any comments on absolute oxygen consumption values in a study combining male and female patients (without having information on percent predicted values) are scientifically not sound.

Response 1: Thank reviewer for the helpful and critical comment. As reviewer indicated, the small number of studies is a main limitation of this study. Also, the information of the percent predicted values for different genders are not estimated in the included studies; accordingly, we determined only the value of VO2peak, as raw data, in this study. In the conclusions, we clarified these as limitations and weak points of this study. Further study should be done in comparisons of male and female in larger scales. We highly appreciate your comment on this.

Reviewer 2 Report

The authors have done a good job of addressing my concerns from the first submission. I do question however why the additional information was not included in the original version of the manuscript as it is all standard inclusions for manuscripts of this type. I want to implore the authors to follow PRISMA guidelines and report search and data extraction strategies in the requisite depth as a default approach, not simply after it being flagged by a reviewer.

The only thing I suggest the authors consider - for Point #11, I did understand the figure, I just believe it will be more intuitive to readers if the directions presented were flipped to indicate that ME/CFS were to the left of the line and therefore scored WORSE. Presenting it (although correctly based on the methods of calculation they have described) with the ME/CFS populations appearing to the right on the forest plot may lead some people to interpret that ME/CFS sufferers perform better on such a test, as data points to the right of the line on forest plots are often used to represent an increase/positive change. No change required if the authors are comfortable, but I suggest it may be more appropriate in terms of data presentation.

Author Response

  1. why the additional information was not included in the original version of the manuscript as it is all standard inclusions for manuscripts of this type.

Response 1: Thank reviewer for the comment. We fully understand your comment and concern on this. Initially, we tried to design a concise manuscript with the supplementary figure and tables, it may have caused the omissions from the main text. We have now carefully re-stated and amended the methodology during this revision process. We highly appreciate your suggestions.

  1. it will be more intuitive to readers if the directions presented were flipped to indicate that ME/CFS were to the left of the line and therefore scored WORSE.

Response 2: Thank reviewer for the helpful comment. As you suggested, we have made amendments for the Table 3 and Figure 2, and positioned the data of ME/CFS patients to the left of the figure as the results of CPET test 2 – 1.